# Study of the Brain Functional Connectivity Processes During Multi-Movement States of the Lower Limbs

**DOI:** 10.3390/s24217016

**Published:** 2024-10-31

**Authors:** Pengna Wei, Tong Chen, Jinhua Zhang, Jiandong Li, Jun Hong, Lin Zhang

**Affiliations:** 1Academy of Air and Missile Defense, Air Force Engineering University, Xi’an 710051, China; pengna_wei@163.com (P.W.); tongch918@163.com (T.C.); long_waver@163.com (L.Z.); 2The Key Laboratory of Education Ministry for Modern Design and Rotor-Bearing System, School of Mechanical Engineering, Xi’an Jiaotong University, Xi’an 710049, China; jjshua@mail.xjtu.edu.cn (J.Z.); jhong@mail.xjtu.edu.cn (J.H.)

**Keywords:** functional connectivity, time–frequency cross-mutual information (TFCMI), electroencephalography (EEG), lower-limb movement

## Abstract

Studies using source localization results have shown that cortical involvement increased in treadmill walking with brain–computer interface (BCI) control. However, the reorganization of cortical functional connectivity in treadmill walking with BCI control is largely unknown. To investigate this, a public dataset, a mobile brain–body imaging dataset recorded during treadmill walking with a brain–computer interface, was used. The electroencephalography (EEG)-coupling strength of the between-region and within-region during the continuous self-determinant movements of lower limbs were analyzed. The time–frequency cross-mutual information (TFCMI) method was used to calculate the coupling strength. The results showed the frontal–occipital connection increased in the gamma and delta bands (the threshold of the edge was >0.05) during walking with BCI, which may be related to the effective communication when subjects adjust their gaits to control the avatar. In walking with BCI control, the results showed theta oscillation within the left-frontal, which may be related to error processing and decision making. We also found that between-region connectivity was suppressed in walking with and without BCI control compared with in standing states. These findings suggest that walking with BCI may accelerate the rehabilitation process for lower limb stroke.

## 1. Introduction

Gait disorders have brought a lot of inconvenience to the lives of many stroke patients. The occurrence of human gait is a complex neurophysiological process that requires the coordination of the cerebral cortex, thalamus, basal ganglia, cerebellum, brainstem, spinal cord, and muscles [1]. Therefore, it is critical to monitor cortical activity during walking. Electroencephalography (EEG) has been widely applied in the analysis of lower limb motion [2,3]. Lau et al. studied the sensorimotor connectivity of walking versus standing and found that the connections were weaker in walking than in standing [4]. Johanna Wagner et al. also found significant differences in cortical activation between active and passive robot-assisted gait [5]. Thomas C. Bulea et al. developed a user-driven treadmill control scheme to study the mental engagement during user-driven treadmills and passive treadmills [6] and found that user-driven treadmills engage the motor cortex more than passive treadmills. However, the cortical functional connectivity between passive and positive treadmill walking is largely unknown. Therefore, cortical functional connectivity during treadmill walking with and without brain–computer interface (BCI) control has to be studied to investigate the neural mechanism for BCI-based lower-limb stroke rehabilitation.

Studies have found that the electrocortical activity during walking is coupled to gait phases [7]. Actually, the idea of movement is first generated in the motor area of the cerebral cortex and transmitted to the brainstem in the form of electrical signals [8]. Then, the motor command is issued by the spinal cord. Subsequently, the conversion from electrical energy to chemical energy and then mechanical energy is realized in the muscles, and movement is thus generated. Therefore, EEG signals can be applied to decode kinematics and surface electromyography (EMG) during walking, as demonstrated in [9]. BCIs functions as a bridge [10] between the brain and a computer to enable users to control robots or a virtual avatar with their thoughts [11,12]. BCIs can also be used to decode the lower limb joint angles from EEG to drive an avatar’s leg [13]; thus, the dynamics of cortical involvement in walking were investigated [14]. EEG-based BCI has been shown to be effective in stroke rehabilitation [15]. Troy M Lau et al. investigated the effective functional connectivity of cortical activity during standing and walking [4]. However, the reorganization of cortical functional connectivity in the continuous movements of lower limbs (including standing before walking, treadmill walking, treadmill walking with BCI control, and standing after walking) is largely unknown. The challenge in such an investigation is that cortical functional connectivity is dynamic and economical and varies according to different tasks [16,17].

According to the theory of functional integration, as one of the basic principles of brain functional organization, the occurrence of any motion requires not just one specific neural region but interaction between several specialized neural regions [18]. Previous researchers had analyzed the functional connectivity thoroughly [19] and found the energy consumption difference between one task and another is quite small [20]. This indicates that our brain maintains almost equal energy consumption in two different functional states but has different integration patterns. Therefore, the reorganization of functional connectivity intends to suppress irrelevant cortical networks and activates the task-related functional networks [21]. This motivates the study of the dynamic changes in the cortical networks during continuous movement.

The first research about the modulation of functional connectivity from the resting to motor state was conducted by Jiang et al. [22]. They evaluated the functional connection between any two regions using Pearson’s correlation coefficient [22]. A coherence analysis method was applied to compute cortical functional connections [23]. However, the coherence method may not be suitable for analyzing nonlinear EEG signals. Nevertheless, the time–frequency cross-mutual information (TFCMI) method provides a way to calculate the nonlinear signals [24]. The calculation process of the TFCMI method includes wavelet transformation and mutual information. Furthermore, the TFCMI analysis is more sensitive to task-related connections than coherence analysis [24].

Hence, we intend to investigate how the cortical functional connectivity changes during continuous movement. We study the functional connectivity during a sequence of lower-limb movement states (including mental-engaged and normal movements) using the TFCMI analysis. This facilitates the investigation of the reorganization of cortical functional connectivity behind continued lower limb movement for both mentally engaged and normal movement.

The rest of our research is organized as follows: Section 2 introduces the datasets and experimental protocols. The EEG signal-processing and brain connectivity analysis using TFCMI are also presented in this part. Section 3 presents the results of the accumulated coupling strength of brain connection in the time–frequency domain and accumulated coupling connectivity difference between two states in the time–frequency domain. BrainNet Viewer 1.0 is used to show the significant region–pair connections. Section 4 discusses the results in depth and the limitations of our research. Section 5 summarizes the main points of our findings.

## 2. Materials and Methods

In this section, we will introduce the datasets and experiments protocol first. Then, the implementation process of EEG-preprocessing methods and TFCMI algorithm will be introduced.

### 2.1. Datasets and Experiments Protocol

We used a public dataset of the University of Houston, namely a mobile brain-body imaging dataset recorded during treadmill walking with a brain–computer interface [13]. The EEG and electrooculography (EOG) data were recorded from eight healthy subjects using 64 64-channel EEG cap (ActiCap system, Brain Products GmbH, Gilching, Germany) with a sample rate of 100 Hz. The 64 channels include 4 eye electrical channels, 1 reference channel, and 1 ground channel. The 58 EEG channel layout is shown in Figure 1a. The bilateral hip, knee, and ankle joint angles were recorded with six goniometers (SG150 and SG110/A Gonio electrodes, Biometrics Ltd., Newport, UK) at 100 Hz [13]. An RF wireless interface (MOVE system, Brain Products GmbH, Gilching, Germany) was used to transmit data to the host PC at its default working band [13]. A trial includes two standing phases, treadmill walking (W), and treadmill walking with closed-loop BCI (WB) phases. The protocol timeline is shown in Figure 1b. Each subject completed three identical trials. A virtual avatar was displayed at eye level on a 52-inch monitor placed in front of the treadmill. At the beginning and end of the trial, each subject stood for two minutes. After the standing phase at the beginning, the subjects walked on the treadmill at 0.45 m/s for two phases: W (15 min) and WB (5 min). In the 15 min of W phases, the subjects walked on a treadmill, and the joint angles of the avatar were controlled by the subjects’ joint angles measured by the goniometers. During the 5 min WB phase, a closed-loop BCI was used to control the avatar. The right leg of the avatar was controlled by the subject’s brain activity, while the left leg was still controlled by the measured joint angles of the subject’s left leg. The detailed experiment protocol can be found in [13]. All experiments were performed in accordance with the 45 Code of Federal Regulations part 46, specifically addressing the protection of human study subjects as promulgated by the U.S. Department of Health and Human Services (DHHS).

### 2.2. EEG Signal Processing and Preparation

The motion artifact can be neglected at lower walking speeds according to [25]. The selected walking speed in this experiment was low enough to minimize motion artifacts. The signal processing was performed in Matlab R2016b and plug-in toolbox EEGLAB (http://www.sccn.ucsd.edu/eeglab/, (accessed on 15 September 2024)) [26]. We excluded the two reference channels and four EOG channels. All the EEG channels were band-pass-filtered at 0.1–50 Hz. Channels with standard deviation > 1000 μV and kurtosis more than 5 standard deviations from the mean were removed [14]. Then, the remaining channels were channel average re-referenced. Subsequently, the EEGLAB tool-box plug-in of Artifact Subspace Reconstruction (ASR) was applied to remove muscle movements and electrode-shift-related artifacts [27]. The ASR algorithm identified the channels with movement artifacts using principal component analysis. Thereafter, the EEG channels were reconstructed using the mixing matrix which is calculated from the baseline data. The channel would be removed if flat for more than 5 seconds. The threshold of channel cross-correlation was set at 0.7. The SD value for repair bursts using ASR was set at 10. The ocular-related components were excluded by the independent component analysis (ICA) method [28]. The ocular correction ICA was conducted in Analyzer 2.0 software, which is developed by Brain Products Company (www.brainproducts.com, (accessed on 15 September 2024)). The ICA algorithm we used was Infomax. We used a data interval with a length of 50 s to calculate the ICA matrix. We also used channel interpolation. The EEG-preprocessing steps are shown in Figure 2.

Next, we split the data into four matrices (standing before treadmill walking: 12,000 × 58 (2 min samples), treadmill walking: 90,000 × 58 (15 min samples), treadmill walking with closed-loop BCI: 30,000 × 58 (5 min samples), standing after treadmill walking with closed-loop BCI: 12,000 × 58 (2 min samples). The rows of the matrix are the EEG samples, and 58 is the number of EEG channels. We selected the same length (5 min) data of W and WB phases for analysis. Then, the EEG data were segmented into a single gait cycle by joint angles (about 180 gait cycles in five minutes of walking). Subsequently, the three identical trials of subjects were appended. Then, the TFCMI was applied to analyze the brain connectivity (including grouping the EEG channels into 10 regions, computation of differences between two states, and statistics on region–pair connectivity).

### 2.3. Brain Connectivity Analysis Using TFCMI

The TFCMI method is then applied to compute the mutual information among different signals [24]. It measures the nonlinear rather than linear information between two signals compared with coherence analysis [29]. The TFCMI algorithm is detailed in [21,29]. Firstly, we select the frequency bands of EEG signals. The delta (0.1–4 Hz) band has been demonstrated to account for over 90% of the total power spectral density (PSD) for the kinematics [30]. The Theta (4–7 Hz) band power has been shown to increase in encoding new information [31]. The low gamma (30–50 Hz) bands had been shown to provide more information related to attention to gait [32]. Therefore, the above three frequency bands were analyzed in this study. All EEG signals were normalized before TFCMI analysis (Figure 3a). Secondly, Morlet wavelet transformation was applied to transform the EEG channels into the time–frequency domain (Figure 3b). The Morlet wavelet transformation is conducted by the Morlet Wavelet function in Matlab. The frequency band of EEG used for analysis was from 16 Hz to 25 Hz. The step of frequency was 1 Hz. The delta, theta, and low gamma band’s time–frequency power maps were obtained by wavelet transformation. Then, the averaged time–frequency power was used to calculate the cross-mutual information between every two EEG channels (Figure 3c).

Finally, the nonlinear correlation between the two channels was computed by mutual information. The cross-mutual information (CMI) between any two channels was calculated using the mean power signals. CMI maps were created by computing the entropy and mutual information, which can be expressed as follows [28]:$$H\left(F_{i}\right)=-\overset{40}{\underset{b=1}{\sum }}p\left(F_{i,b}\right)\log_{2}p\left(F_{i,b}\right)$$
$$TFCMI\left(F_{i},F_{j}\right)=H\left(F_{i}\right)+H\left(F_{j}\right)-H\left(F_{i},F_{j}\right)=\overset{40}{\underset{b=1}{\sum }}p\left(F_{i,b},F_{j,b}\right)\log_{2}\frac{p\left(F_{i,b},F_{j,b}\right)}{p\left(F_{i,b}\right)p\left(F_{j,b}\right)}$$
where $H\left(F_{i}\right)$ denotes entropy, $F_{i}$ is mean power signals at the ith channel, $p\left(F_{i,b}\right)$ is the probability density function (PDF) of $F_{i}$, $p\left(F_{i,b},F_{j,b}\right)$ is the joint probability density function (JPDF) of $F_{i},$ while $F_{j}$, b=1,2,⋯,40 is the bin number of the histogram used to construct the approximated PDF. Some 40 bins were selected based on both previous research and our data [30]. The TFCMI values are then represented as an asymmetric 58 × 58 TFCMI map (Figure 3d). The accumulated coupling strength was calculated by summing up the rows or columns of the 58 × 58 TFCMI map (Figure 3e). For example, the obtained 1 × 58 vector represents the contribution of one channel to all 58 channels.

### 2.4. Computation of Differences Between Two States

If we consider the pairwise connectivity of accumulated coupling strength, 58 channels of EEG signals will result in $1653=\frac{\left(58\times 58-58\right)}{2}$ pairwise connections, minus 58 if self-to-self connections were excluded. A large number of pairwise connections will yield results that are hard to interpret. Therefore, the 58 channels were grouped into 10 regions [21]: RF, LF, MF, RC, LC, MC, RP, LP, MP, and O, as detailed in Table 1. We exclude some channels that are contaminated by neck muscles and not closely related to gait [28]. We then create a 10 × 10 TFCMI map by averaging the TFCMI values within any pairwise connections of all divided regions. The selected channels are displayed in Table 1.

The squared biserial correlation coefficient is a statistical way to compute the difference between any two different states. The calculation is as follows [28]:$$Y_{r}\left(m,n\right)=\frac{\sqrt{N_{s1}\cdot N_{s2}}}{N_{s1}+N_{s2}}\cdot \frac{mean\left(TFCMI_{s1,a}\left(m,n\right)\right)-mean\left(TFCMI_{s2,b}\left(m,n\right)\right)}{std\left(TFCMI_{s1,a}\left(m,n\right)\cup TFCMI_{s2,b}\left(m,n\right)\right)}$$
where *m* = 1, 2, 3, …, 10, *n* = 1, 2, 3, …, 10, *a* = 1, 2, 3, …, $N_{s1}$, *b* = 1, 2, 3, …, $N_{s2}$. $N_{s1}$ and $N_{s2}$ represent the number of trials of states s1 and s2. The states include standing before walking, treadmill walking, treadmill walking with closed-loop BCI control, and standing after walking. The larger the value of $Y_{r}\left(m,n\right)$ is the greater difference between the two states in the channels *m* and *n*. The same as TFCMI maps, when we sum up the matrix $Y_{r}$ along the rows or columns, the coupling difference of a single channel to the global network (10-to-1 connectivity) is obtained. Accumulated coupling connectivity difference BrainNet Viewer between two states obtained by $Y_{r}\left(m,n\right)$. An independent-samples Kruskal–Wallis test was used to analyze the statistical difference in SPSS 23.0 (IBM Corp., Chicago, IL, USA).

## 3. Results

In the following, the accumulated coupling strength of the brain connection is calculated by TFCMI and shown in topographic maps. The significant region–pair connection is calculated and shown by BrainNet Viewer 1.0.

### 3.1. Accumulated Coupling Strength of Brain Connection in the Time–Frequency Domain

The topographic maps of the averaged accumulated coupling strength were presented in Figure 4 (a) delta, (b) theta, and (c) gamma bands. In each frequency band, we displayed the four states. The EEG channels at the edge were not displayed in the topographic maps, which has been explained in Section 2.4. First, we summed up along the rows of the TFCMI matrix of each subject. Then, we averaged the coupling strength of all subjects.

To improve the readability of the figure, Aft represents the standing after WB, and Pre represents the standing before W. Figure 4a showed that for the delta band, the accumulated coupling strength of the occipital area was lower than other cortical areas. After WB, there appeared to be an increase in accumulated coupling strength of the frontal and central areas. Figure 4b showed that for theta band, the accumulated coupling strength of the occipital area was decreased at standing after the WB state. Figure 4c shows that for the gamma band compared with the W state, the accumulated coupling strength decreased in the frontal and central areas during the WB state.

### 3.2. Accumulated Coupling Connectivity Difference Between Two States in the Time–Frequency Domain

Figure 5 shows the accumulated coupling connectivity strength changes in 10 regions between two functional states. The correlation coefficient between two states is computed in the previous section using a statistical method. A larger correlation coefficient value shows a larger contribution to the change in the global network. The results showed that the modulations of contribution from the state of standing before W to W decreased in all regions except the right-parietal area. And the accumulated coupling connectivity of the left-frontal area is down to the minimum. The absolute modulations from W to WB in the middle parietal area are the largest (*p* < 0.001).

Furthermore, the modulations are decreased in all regions except the left-frontal area. Figure 5a shows that in the two standing states, the correlation coefficients also decreased from standing before W to standing after WB in all regions except the occipital area. For the theta band, the modulations of contribution from the state of standing before W to W are decreased in all regions except a slight increase in left-parietal and occipital areas, and especially the left- and middle-frontal areas decreased obviously. In two walking states, the modulations of contribution from the W to WB increased sharply in the left-frontal area (*p* = 0.049). Figure 5b shows that in two standing states, the modulations of contribution from the state of standing before W to standing after WB are increased obviously in the left-parietal and -occipital areas. For the gamma band, the decreases in modulations of contribution from the state of standing before W to W are primarily distributed in the areas of occipital, left- and right-parietal, left- and right-central, and left-frontal. On the contrary, the modulations of contribution from W to WB show an obvious increase in all regions (*p* < 0.001). There is no obvious law in the modulations of contribution from the states of standing before W to standing after WB. Such as, it increases in the right- and middle-frontal, and right-central areas, and decreases in the left-frontal, left-central, left-parietal, and -occipital areas. Figure 5c also shows no change in contribution modulations in other areas.

### 3.3. Statistic on Region–Pair Connectivity

We have analyzed the accumulated coupling connectivity strength changes in 10 regions between two functional states. We use nodes to represent the connectivity strength within regions and edges for the connectivity strength between two regions. The nodes are the accumulated coupling strength, which is calculated by summing up along rows of the 10 × 10 TFCMI maps described in Section 2.4. The edges represent the *p* value of region–pair connectivity. The Mann–Whitney U-test and multiple comparisons are applied to analyze whether the difference in region–pair connectivity was significant. After creating the difference matrix *p* values of the four states, we adopted BrainNet Viewer to present the results, and the significant region–pair connection is shown in Figure 5 (*p* > 0.05). BrainNet Viewer was developed by Mingrui, Xia et al., which is a visual network analysis toolbox [33].

Figure 6 shows that most of the between-regions are not connected significantly for the two standing states for the delta, theta, and gamma bands. Figure 6a shows that the *p* value of region–pair connectivity between left-parietal and -occipital areas is the largest in W and WB for the delta band. The connectivity strength within regions is stronger in the central cortical area than in the left and right areas. Figure 6b shows that the *p*-value between the left-central and middle parietal areas is large both in W and WB for the theta band. Furthermore, the *p* values of multiple comparisons between the left and right cerebral hemispheres are also >0.05 in the theta band, especially the region–pair of the right frontal and the left-central areas, the right-parietal and left-frontal areas. Figure 6c shows that the *p*-value between the right cerebral hemisphere and middle areas is large in both W and WB for the gamma band (right-central and middle-parietal areas for W, right-central and middle frontal areas for WB).

## 4. Discussion

In this study, we first investigated the dynamic changes in cortical functional connection during four movement states. We discovered that the frontal–occipital connection increased in walking with BCI states in Figure 6c, which showed effective communication when subjects adjusted their gait to control the avatar. Furthermore, the functional connectivity of walking irrelated regions was suppressed during walking and walking with BCI control states compared with standing states. This suggests that the functional network system is economical and focused on the task-related cortices [34] during lower-limb movement. We also found a significant increase in within-region connectivity in frontal, central, parietal, and occipital cortices in the walking-with-BCI control compared with the walking state in the gamma band in Figure 5c, indicating increased attention and active control of gait in walking with BCI control [32]. The significant decrease in contribution to global networks in the left-frontal can be observed from the W to W + BCI states in Figure 5b. The averaged accumulated coupling strength was decreased in the occipital area of standing after W + BCI state in the delta band but increased in frontal and central cortices in Figure 4a. This may be due to the subjects’ relaxed states during the experiments of standing before walking, walking, and walking with BCI control. However, the subjects paid attention to the instructions after walking with BCI control, and the accumulated coupling strength was decreased in the occipital and increased in the frontal and central cortices to maintain a steady energy consumption. The same phenomenon can be observed in the theta band, which is consistent with previous research [20].

The modulations of contribution from the state of standing before W to W almost decrease in all regions, which can be observed in Figure 5a–c. This indicated that the contributions of most regions decreased in motion compared with the standing state. Furthermore, in the two standing states, the modulations of the contribution of the delta band decreased in all regions except occipital cortices. This may be due to the delta band accounting for more than 90% of the total PSD for the kinematics [30]. Compared with standing before walking, the frontal–central–parietal network was deactivated for the preparation of the next task in standing after walking.

For the theta band, the modulations of contribution from the W to WB are significantly increased in the left-frontal area (*p* = 0.049) as shown in Figure 5b. Previous research has demonstrated that theta oscillations of EEG signals correlate with task difficulty [35], error-monitoring and -learning processes [36], memory, and decision-making [37]. Theta activity increment in the frontal cortex and hippocampus has been proved during cognitive control demands [31]. This indicates that the subjects attempt to adjust their gait to ensure that the avatar’s right leg can be consistent with their own. In the gamma band, the modulations of contribution from W to WB show a significant increase in all regions (*p* < 0.001) in Figure 4c. Martin et al. proved that the gait phase-related modulations in the low gamma represent the motion sequence timing during gait [38]. Alvaro et al. also discovered that the gamma band provides more information related to attention during gait [32]. According to the mentioned findings, the results suggest that the subjects try to adjust their gait to control the avatar during walking with BCI control state. Therefore, BCI-based rehabilitation has the potential to increase the mental engagement of stroke patients.

The connectivity network in TFCMI values is shown in Figure 6. The connectivity network of the two walking states was suppressed compared with the two standing states. This can be explained by the economy of the connectivity network. The between-region connectivity in two standing states was extensive; then, the walking irrelevant cortical networks were suppressed when the state switched to walking. The between-region connectivity of left hemisphere regions was strengthened during walking with BCI control state compared with the walking state, such as the between-region connectivity between left-frontal and left-parietal areas and the left-frontal and -occipital areas, which can be observed in Figure 6a. On the contrary, the between-region connectivity of right-hemisphere regions was suppressed during walking with BCI control states, such as the between-region connectivity between the right-frontal and right-central areas and the right-parietal and -occipital areas disappeared. The dynamic changes in connectivity networks from left to right regions when the right leg of the avatar was controlled by decoded joint angles. The most plausible explanation is that the active adjustment of gait requires the contralateral frontal–occipital networks to control the joint angle. This is consistent also with our previous conclusion.

The limitations of our study are as follows: Firstly, we applied our analysis to available data and did not conduct our experiments. We intend to analyze more daily movements of the lower limb in our future works, such as climbing up and down stairs, walking uphill and downhill, and so forth. Secondly, source localization is also a vital method in EEG signals analysis, but we just analyzed the functional networks in our current work. We will analyze the independent component source clusters during different daily movements in the future. Finally, the sample size of the dataset in our study is relatively small. A study with a large cohort of subjects is needed.

## 5. Conclusions

In this study, we focused on the dynamic modulation of network functional connectivity during continuous limb movements. The two main findings are as follows: Firstly, the frontal–occipital connection increased in gamma and delta during walking with BCI states showing effective communication when subjects adjust their gait to control the avatar. This indicates the BCI-based rehabilitation of stroke increased motor learning ability and mental engagement. Secondly, the functional connectivity of walking irrelated regions was suppressed during walking and walking with BCI control states compared with standing states. Thirdly, compared with standing before walking, the frontal–central–parietal network was deactivated for the preparation of the next task in standing after walking. These findings showed that walking with BCI increases motor learning ability.

## Figures and Tables

**Figure 1 sensors-24-07016-f001:**
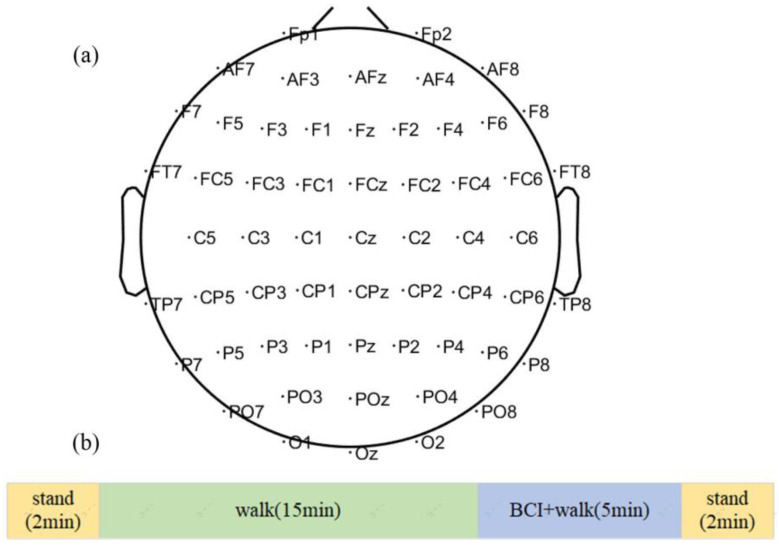
Experimental paradigm: (**a**) EEG channel layout and (**b**) protocol timeline.

**Figure 2 sensors-24-07016-f002:**
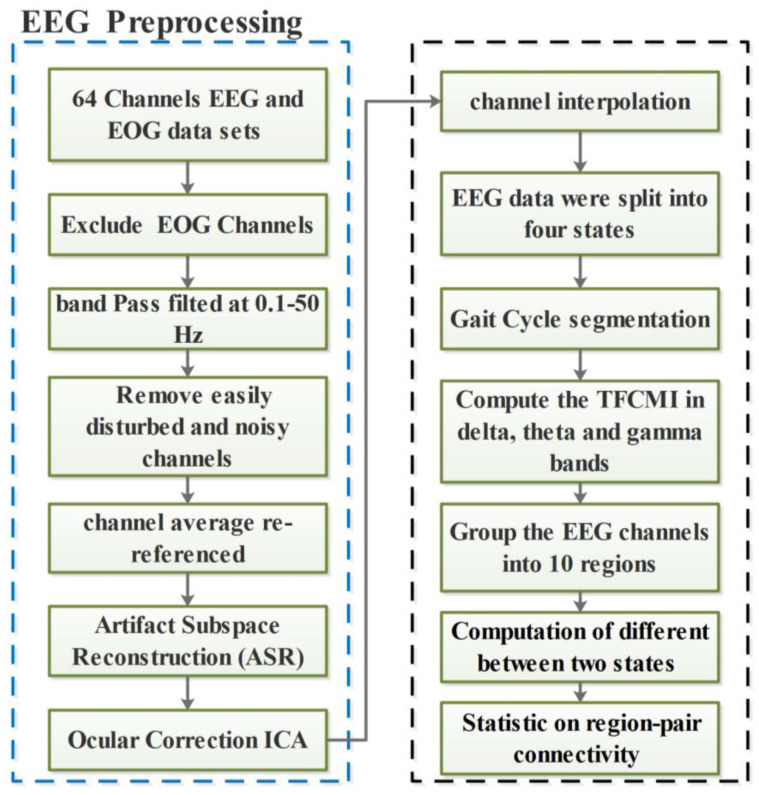
EEG-preprocessing steps.

**Figure 3 sensors-24-07016-f003:**
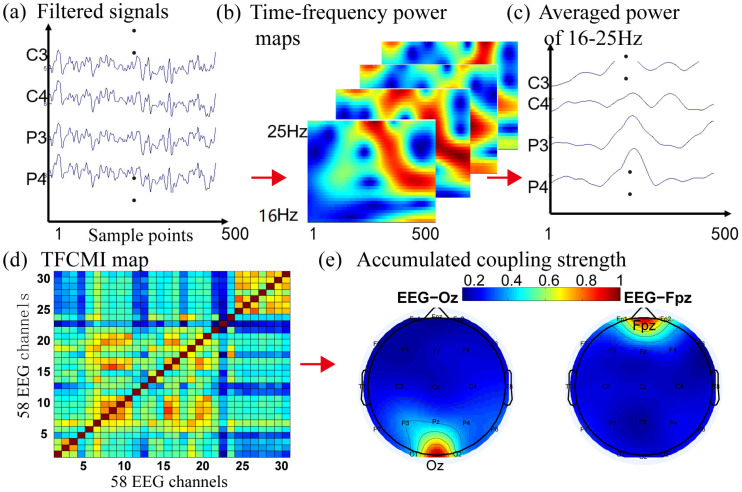
The calculation process of TFCMI. (**a**) The raw EEG obtained from 58 channels was first filtered by a bandpass filter (with 0.1–50Hz passband). (**b**) The filtered signals of each channel were processed using the Morlet wavelet transformation to obtain time–frequency power maps within the selected frequency band (16–25 Hz). (**c**) The averaged power signal for each channel was created by averaging the individual time–frequency maps across the selected frequency band. (**d**) The 58 × 58 TFCMI map was obtained by calculating the TFCMI values from the averaged powers between any two channels. (**e**) The accumulated coupling strengths can be represented by summing the rows or columns of TFCMI maps and depicted as a 58-channel topographic map.

**Figure 4 sensors-24-07016-f004:**
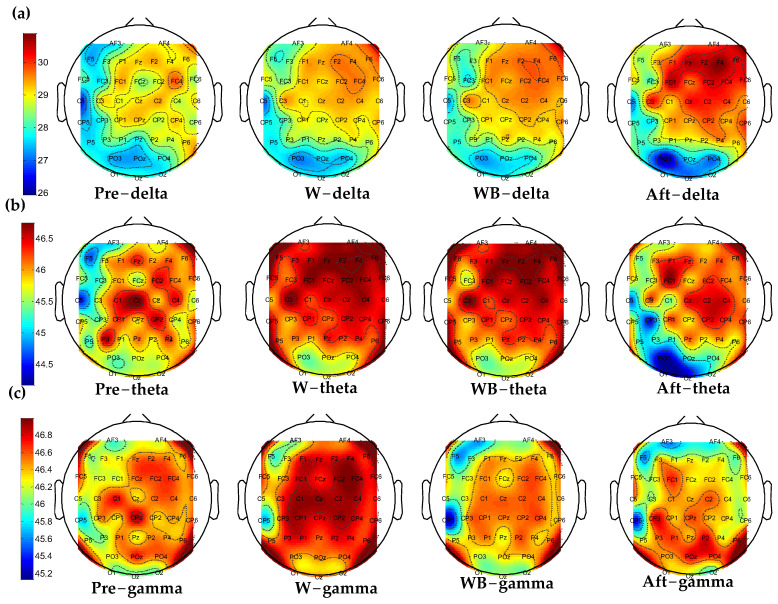
The topographic maps of the averaged accumulated coupling strength of eight subjects: (**a**) delta, (**b**) theta, and (**c**) gamma bands. Aft is the standing after W + BCI; Pre is the standing before W.

**Figure 5 sensors-24-07016-f005:**
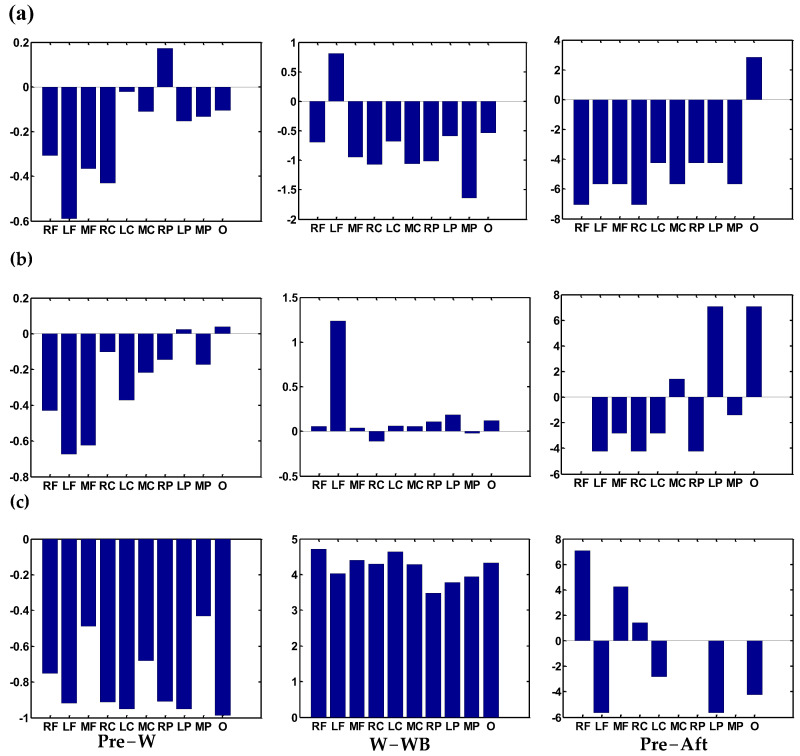
The accumulated coupling connectivity difference between two states, including the modulation of 10-to-1 connectivity from Pre to W, W to WB, and Pre to Aft: (**a**) delta band, (**b**) theta band, (**c**) gamma band. Aft is the standing after W + BCI; Pre is the standing before W.

**Figure 6 sensors-24-07016-f006:**
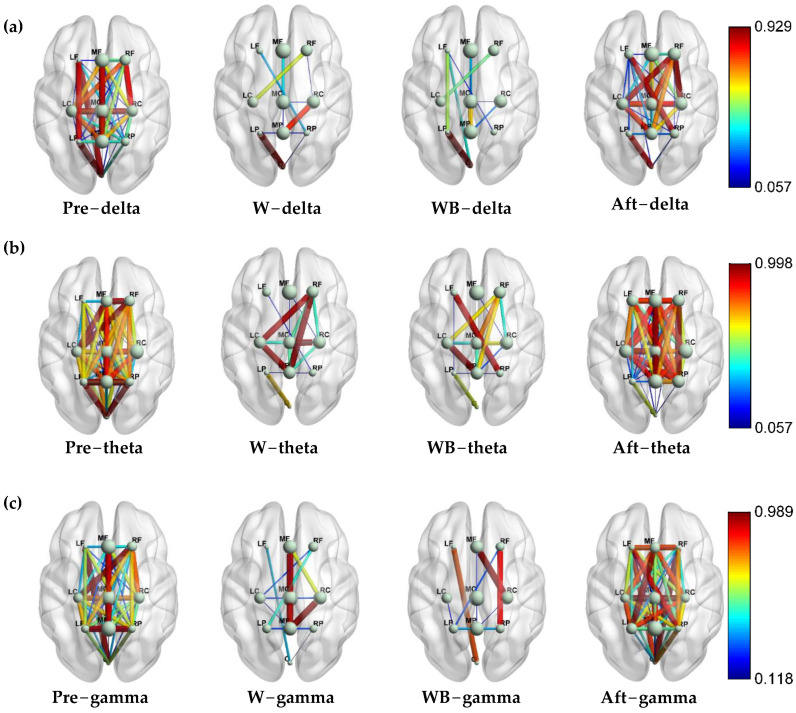
The statistical analysis of the connectivity network in TFCMI values for the four states: (**a**) delta, (**b**) theta, and (**c**) gamma-band; the green balls represent the within-region connectivity, and the lines are the significant between-region connectivity.

**Table 1 sensors-24-07016-t001:** The divided regions and channels of each region.

Label	Cortical	Channels
RF	Right Frontal area	F2, F4, FC2, FC4
LF	Left-Frontal area	F1, F3, FC1, FC3
MF	Middle Frontal area	Fz, FCz
RC	Right Central area	C2, C4, FC2, CP2
LC	Left Central area	C1, C3, CP1, FC1
MC	Middle Central area	Cz, CPz, FCz
RP	Right Parietal area	CP2, CP4, P2, P4
LP	Left-Parietal area	CP1, CP3, P1, P3
MP	Middle Parietal area	CPz, Pz
O	Occipital	PO3, PO4, POz, O1, O2, Oz

## Data Availability

We used a public dataset of the University of Houston. https://doi.org/10.6084/m9.figshare.c.3894013 (accessed on 15 September 2024) (2018).

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
