# Peer review of "Study of the Brain Functional Connectivity Processes During Multi-Movement States of the Lower Limbs"

_sensors, 2024, doi:10.3390/s24217016_

Round 1

Reviewer 1 Report

Comments and Suggestions for Authors

This study explores the reorganization of cortical functional connectivity during treadmill walking with BCI control through the analysis of EEG coupling strength. The research results reveal that, during BCI walking, the frontal-occipital connection in the gamma and delta bands increases, possibly related to effective communication when subjects adjust their gaits to control the avatar. Additionally, theta oscillations within the left frontal region may be associated with error processing and decision-making. 

The authors are encouraged to explore the following points further:

1. Provide more details on the experimental design and data analysis methods to improve the study's reproducibility.

2. Consider comparing the effects of different types of BCI control modes on cortical functional connectivity.

3. Investigate the differential impacts of BCI walking on various populations, such as healthy individuals versus stroke patients.

4. The authors did not discuss how electrode shifts during movement affect classification performance, although this impact seems significant for BCI under motion conditions.

5. The manuscript includes a topographic map representing EEG data, but it appears that the map displays approximately 43 channels instead of the expected 58 channels. This discrepancy raises concerns about whether the topographic map has been accurately represented or if there was an error in its creation.

6. In Table 1, the manuscript lists 10 regions with their corresponding EEG channels. However, there appears to be a lack of citation or literature support for the selection of these specific channels for each region.

7. The results section of the manuscript would benefit from a more in-depth discussion. Currently, it seems to primarily analyze results that are visually apparent from the figures, without delving into deeper insights or interpretations. Additionally, there are errors in some table captions, and it is unclear whether the authors have validated their findings using fMRI data.

Comments on the Quality of English Language

No.

Author Response

  • Comments 1:

Provide more details on the experimental design and data analysis methods to improve the study's reproducibility.

Responses to the Comments 1:

We appreciate the valuable suggestion that greatly improves the quality of the manuscript. We have carefully checked the experimental design and data analysis parts. And provided more information about EEG electrodes and experimental protocol. All the changes in the Materials and Methods have been highlighted in yellow and green.

  • Comments 2:

Consider comparing the effects of different types of BCI control modes on cortical functional connectivity.

Responses to the Comments 2:

Thanks for the constructive comments, but this study focuses on investigating the brain functional networks of four different states. Your suggestion will be one of the important topics for our future research.

  • Comments 3:

Investigate the differential impacts of BCI walking on various populations, such as healthy individuals versus stroke patients.

Responses to the Comments 3:

Thanks for the constructive comments. However, the reorganization of cortical functional connectivity in treadmill walking with BCI control is largely unknow in healthy individuals, let alone stroke patients. We need to investigate the dynamic changes in brain functional networks during continuous walking with BCI and without BCI. Thank you for your suggestion. We are considering trying your proposal next.

  • Comments 4:

The authors did not discuss how electrode shifts during movement affect classification performance, although this impact seems significant for BCI under motion conditions.

Responses to the Comments 4:

Thank you very much for your comment. Motion artifacts are negligible at lower speeds. In the experiment, the subjects walked at a lower speed. We also used ASR to remove the muscle movements and electrode shifts related artifacts.

  • Comments 5:

The manuscript includes a topographic map representing EEG data, but it appears that the map displays approximately 43 channels instead of the expected 58 channels. This discrepancy raises concerns about whether the topographic map has been accurately represented or if there was an error in its creation.

Responses to the Comments 5:

Thank you very much for carefully reviewing our initial manuscript. We exclude some channels that are contaminated by neck muscles and not closely related to gait. Therefore, the map displays 43 channels instead of 58 channels.

  • Comments 6:

In Table 1, the manuscript lists 10 regions with their corresponding EEG channels. However, there appears to be a lack of citation or literature support for the selection of these specific channels for each region.

Responses to the Comments 6:

We thank the reviewer for the kind comment on our manuscript. We have cited the referenced literature.

  • Comments 7:

The results section of the manuscript would benefit from a more in-depth discussion. Currently, it seems to primarily analyze results that are visually apparent from the figures, without delving into deeper insights or interpretations. Additionally, there are errors in some table captions, and it is unclear whether the authors have validated their findings using fMRI data.

Responses to the Comments 7:

Thank you very much for your comment. We conducted a thorough analysis of the experimental results in the discussion section. Thank you very much for carefully reviewing our manuscript. We have corrected this error. All the changes have been highlighted in yellow. We did not validate the findings using fMRI data.

Reviewer 2 Report

Comments and Suggestions for Authors

The study is interesting, but some clarifications are needed. The following comments are provided for improvement:

  • In the Abstract section, it is unclear what metric was used for evaluation. Additionally, it should be specified how the gamma and delta alterations were significant.
  • Please include information about the dataset in the Abstract section.
  • The Introduction section is not strong enough. Consider incorporating more references to existing studies.
  • Since the study focuses on BCI applications and pattern identification methods, it is recommended to cite recent review papers on BCIs, such as:
    • Hekmatmanesh A, Nardelli PH, Handroos H. Review of the state-of-the-art of brain-controlled vehicles. IEEE Access. 2021 Jul 27;9:110173-93.
    • Moioli RC, Nardelli PH, Barros MT, et al. Neurosciences and wireless networks: The potential of brain-type communications and their applications. IEEE Communications Surveys & Tutorials. 2021 Jun 23;23(3):1599-621.
  • The final paragraph of the Introduction should highlight the authors' contributions. Additionally, it is advisable to outline the next steps of the research in this section.
  • The sections '2. Materials and Methods' and '3. Results' are currently empty. Please provide an overview of the content that will be presented in these subsections.
  • Some parts should be divided into multiple paragraphs for better readability. For example, sections 3.1 and 3.2 are presented in single paragraphs.
  • Lastly, please add a 'Conclusion' section after the Discussion to summarize the key findings of the study.
Comments on the Quality of English Language

The language required to be polished.

Author Response

  • Comments 1:

In the Abstract section, it is unclear what metric was used for evaluation. Additionally, it should be specified how the gamma and delta alterations were significant.

Responses to the Comments 1:

We appreciate the valuable suggestion that greatly improves the quality of the manuscript. We have added evaluation methods in the Abstract. We also specified the reason for the significant gamma and delta in the Abstract. All the changes have been highlighted in turquoise.

  • Comments 2:

Please include information about the dataset in the Abstract section.

Responses to the Comments 2:

Thanks for the constructive comments. We have added information about the dataset in the Abstract and have been highlighted in turquoise.  

  • Comments 3:

The Introduction section is not strong enough. Consider incorporating more references to existing studies.

Responses to the Comments 3:

Thanks for the constructive comments. we have revised the Introduction section. All the changes have been highlighted in turquoise and green.

  • Comments 4:

Since the study focuses on BCI applications and pattern identification methods, it is recommended to cite recent review papers on BCIs, such as:

  • Hekmatmanesh A, Nardelli PH, Handroos H. Review of the state-of-the-art of brain-controlled vehicles. IEEE Access. 2021 Jul 27;9:110173-93.
  • Moioli RC, Nardelli PH, Barros MT, et al. Neurosciences and wireless networks: The potential of brain-type communications and their applications. IEEE Communications Surveys & Tutorials. 2021 Jun 23;23(3):1599-621.

Responses to the Comments 4:

Thank you very much for your comment. We have cited recent review papers, including the two you recommended.

  • Comments 5:

The final paragraph of the Introduction should highlight the authors' contributions. Additionally, it is advisable to outline the next steps of the research in this section.

Responses to the Comments 5:

Thank you very much for carefully reviewing our initial manuscript. We have revised the Introduction part, and outline the next steps of the research. All the changes have been highlighted in turquoise.

  • Comments 6:

The sections '2. Materials and Methods' and '3. Results' are currently empty. Please provide an overview of the content that will be presented in these subsections.

Responses to the Comments 6:

We thank the reviewer for the kind comment on our manuscript. We have provided an overview of the content for section 2 and 3. All the changes have been highlighted in turquoise.

  • Comments 7:

Some parts should be divided into multiple paragraphs for better readability. For example, sections 3.1 and 3.2 are presented in single paragraphs.

Responses to the Comments 7:

Thank you very much for your comment. The paragraphs have been revised.

Comments 8:

Lastly, please add a 'Conclusion' section after the Discussion to summarize the key findings of the study.

Responses to the Comments 8:

Thank you very much for your comment. We have added a Conclusion section and have been highlighted in turquoise.

Reviewer 3 Report

Comments and Suggestions for Authors

This study investigated the functional connectivity of EEG during lower-limb movements and concluded that walking with BCI control leads to significant reorganization of cortical connectivity. The findings suggest that BCI-based rehabilitation can increase motor learning and mental engagement, making a valuable contribution to the field. However, the manuscript needs extensive revisions.

The introduction currently feels abrupt, as it reviews previous research without smoothly transitioning into your study's own research questions or objectives. I believe a thorough revision is needed to better frame the context of your research.

For the first paragraph, consider providing a broader background that sets the context for the topic, rather than listing previous studies without a clear rationale.

Similarly, in the second paragraph, the transition to discussing BCIs and their applications seems premature, as it just listed lower-limb applications without establishing the foundational concept of general decoding motor intentions—an essential precursor to understanding motor control. To create a smoother and more scientifically accurate flow, I suggest beginning the second paragraph by discussing how motor intentions can be decoded from neural signals, which is crucial for understanding how movement is planned and executed. Including references that demonstrate how kinematic features are represented during motor planning would also help substantiate this argument, such as [1]. This approach would set the stage for a more natural progression into the role of BCIs in leveraging these decoded intentions for practical applications, ultimately making the paragraph more cohesive and logically structured. From there, you can smoothly transition into your preferred focus on lower limb applications for a more explicit discussion of BCI use in this context.

[1] https://doi.org/10.3389/fnins.2019.01148

The last paragraph of the introduction should clearly present the specific research objectives or hypotheses of the study. While the current paragraph provides a rationale for examining cortical connectivity during lower limb movements, it doesn’t explicitly state what the study aims to accomplish. I recommend concluding the introduction with a concise statement that outlines the study’s primary objective, followed by a brief description of the expected outcomes or hypotheses. This will help establish a clear direction for the reader and provide a stronger foundation for the research that follows.

Crucial technical details are also missing.

List all EEG channels.

Why didn’t you provide a reference for using EEGLAB? Some readers wouldn’t replicate the study if you don’t provide it. Plus, please respect the developers.

The preprocessing steps must be fully disclosed with relevant parameters.

For ASR, what plug-in did you use? Also, the authors didn’t reveal any setting parameters. You must disclose full details.

The preprocessing steps described doesn’t match the flow chart in Figure 1. Describe every step involved, including any omitted steps.

Plus, if you removed noisy channel, didn’t you use channel interpolation? I think you omitted this too.  As I said, please include all the steps.

What ICA algorithm did you use? With what parameter? What tool did you use for ICA?

According to the EEGLAB official documentation, it is crucial to ensure that the data does not have rank deficiency before performing ICA. Please visit the documentation. The documentation highlights that incorrect re-referencing can cause rank deficiency and provides solutions to address this issue. Please explicitly mention this issue and explain how the authors addressed it in detail. I believe it’s likely that you used the PCA option, then you can just say “we performed ICA with the pca option to match the data rank to handle the rank-deficiency issue [a relevant paper]”, or you can say “We re-referenced to the average with the initial reference [a relevant paper], ensuring the full rank of the data”, as recommended including the initial reference for re-referencing.

https://eeglab.org/tutorials/06_RejectArtifacts/RunICA.html#issues-with-data-rank-deficiencies

https://sccn.ucsd.edu/wiki/Makoto's_preprocessing_pipeline#A_study_on_the_ghost_IC_was_published_.28added_on_04.2F04.2F2023.29

Specify all the details of wavelet parameters.

The method section lacks the description of statistical test too.

The last paragraph of the discussion would benefit from a more conclusive statement that clearly summarizes the main findings and their implications. While the current ending highlights some aspects of the results, it does not provide a strong, unifying conclusion that ties everything together. I recommend adding a brief statement that restates the study’s contributions and discusses its broader impact, ensuring that the discussion ends on a clear and impactful note.

Comments on the Quality of English Language

The manuscript should be proofread by the native speaker (e.g., “Electroencephalography (EEG) have been widely studied in lower limb motion analysis”,…)

Author Response

  • Comments 1:

The introduction currently feels abrupt, as it reviews previous research without smoothly transitioning into your study's own research questions or objectives. I believe a thorough revision is needed to better frame the context of your research.

For the first paragraph, consider providing a broader background that sets the context for the topic, rather than listing previous studies without a clear rationale.

Responses to the Comments 1:

We appreciate the valuable suggestion that greatly improves the quality of the manuscript. We have provided a broader background in the first paragraph of the Introduction. All the changes have been highlighted in green.

  • Comments 2:

Similarly, in the second paragraph, the transition to discussing BCIs and their applications seems premature, as it just listed lower-limb applications without establishing the foundational concept of general decoding motor intentions—an essential precursor to understanding motor control. To create a smoother and more scientifically accurate flow, I suggest beginning the second paragraph by discussing how motor intentions can be decoded from neural signals, which is crucial for understanding how movement is planned and executed. Including references that demonstrate how kinematic features are represented during motor planning would also help substantiate this argument, such as [1]. This approach would set the stage for a more natural progression into the role of BCIs in leveraging these decoded intentions for practical applications, ultimately making the paragraph more cohesive and logically structured. From there, you can smoothly transition into your preferred focus on lower limb applications for a more explicit discussion of BCI use in this context.

[1] https://doi.org/10.3389/fnins.2019.01148.

Responses to the Comments 2:

Thanks for the constructive comments. We have revised the second paragraph and explained why motor intentions can be decoded from EEG signals. All the changes have been highlighted in green.

  • Comments 3:

The last paragraph of the introduction should clearly present the specific research objectives or hypotheses of the study. While the current paragraph provides a rationale for examining cortical connectivity during lower limb movements, it doesn’t explicitly state what the study aims to accomplish. I recommend concluding the introduction with a concise statement that outlines the study’s primary objective, followed by a brief description of the expected outcomes or hypotheses. This will help establish a clear direction for the reader and provide a stronger foundation for the research that follows.

Responses to the Comments 3:

Thanks for the constructive comments. We have clearly presented the hypotheses and expected outcomes of our study in the last two paragraph of the introduction. All the changes have been highlighted in turquoise.

  • Comments 4:

Crucial technical details are also missing.

List all EEG channels.

Responses to the Comments 4:

Thank you very much for your comment. We have presented all the EEG channels (exclude EOG, GND and REF channels) in section 2.

  • Comments 5:

Why didn’t you provide a reference for using EEGLAB? Some readers wouldn’t replicate the study if you don’t provide it. Plus, please respect the developers.

Responses to the Comments 5:

Thank you very much for carefully reviewing our initial manuscript. We have provided a reference for using EEGLAB.

  • Comments 6:

The preprocessing steps must be fully disclosed with relevant parameters.

For ASR, what plug-in did you use? Also, the authors didn’t reveal any setting parameters. You must disclose full details.

The preprocessing steps described doesn’t match the flow chart in Figure 1. Describe every step involved, including any omitted steps.

Plus, if you removed noisy channel, didn’t you use channel interpolation? I think you omitted this too. As I said, please include all the steps.

What ICA algorithm did you use? With what parameter? What tool did you use for ICA?

Responses to the Comments 6:

We thank the reviewer for the kind comment on our manuscript.

  • The EEGLAB tool-box plug-in of Artifact Subspace Reconstruction (ASR) was used to conduct ASR. The setting parameters have been presented in section 2.2.
  • We have revised the Figure 2 and describe every step involved. Yes, we used channel interpolation after removing noisy channels. All the changes have been highlighted in green.
  • The ocular correction ICA was conducted in Analyzer software, which is developed by brain products company. The ICA algorithm we used is Infomax. All the changes have been highlighted in green.

  • Comments 7:

Specify all the details of wavelet parameters.

The method section lacks the description of statistical test too.

Responses to the Comments 7:

Thank you very much for your comment. The Morlet wavelet transformation is conducted by Morlet Wavelet function in Matlab. The frequency band of EEG used for analyzing is from 16 Hz to 25 Hz. The step of frequency is 1 Hz. The details of wavelet parameters have been presented in section 2.3.

We have described the statistical test method in section 2.

  • Comments 8:

The last paragraph of the discussion would benefit from a more conclusive statement that clearly summarizes the main findings and their implications. While the current ending highlights some aspects of the results, it does not provide a strong, unifying conclusion that ties everything together. I recommend adding a brief statement that restates the study’s contributions and discusses its broader impact, ensuring that the discussion ends on a clear and impactful note.

Responses to the Comments 8:

Thank you very much for your comment. We have added a Conclusion section and stated the study’s contributions and discusses its broader impact. The changes have been highlighted in turquoise.

  • Comments 9:

Comments on the Quality of English Language

The manuscript should be proofread by the native speaker (e.g., “Electroencephalography (EEG) have been widely studied in lower limb motion analysis”,…).

Responses to the Comments 9:

Thank you very much for your comment. We have proofread the manuscript and revised the syntax error.

Round 2

Reviewer 1 Report

Comments and Suggestions for Authors

The author has comprehensively and thoroughly addressed all the questions I previously raised, fully clarifying the key points and methods of the research. After careful review, I believe the paper's content is rigorous and the research findings possess significant academic value, meeting the standards for publication. Thank you to the author for their hard work and proactive responses. At this time, I have no further questions to raise.

Comments on the Quality of English Language

Aside from a few minor errors in the English language, there are no significant issues.

Author Response

Thank you for your high praise and for taking the time to review our manuscript.

Reviewer 2 Report

Comments and Suggestions for Authors

I am satisfied with the modifications.

Comments on the Quality of English Language

An English language polishing is recommended.

Author Response

Thank you for your high praise and for taking the time to review our manuscript. We will continuously improve our English writing skills.

Reviewer 3 Report

Comments and Suggestions for Authors

I appreciate the authors' sincere response. However, I believe the manuscript still requires improvements, particularly regarding issues that were not addressed in the previous round.

Responses to the Comments 2:

Thanks for the constructive comments. We have revised the second paragraph and explained why motor intentions can be decoded from EEG signals. All the changes have been highlighted in green.

ð  The way the authors cited [9] might be somewhat misleading. I recommend revising it to: "Therefore, EEG signals can be applied to decode kinematics and surface electromyography (EMG) during walking, as demonstrated in [9]."

Responses to the Comments 7:

Thank you very much for your comment. The Morlet wavelet transformation is conducted by Morlet Wavelet function in Matlab. The frequency band of EEG used for analyzing is from 16 Hz to 25 Hz. The step of frequency is 1 Hz. The details of wavelet parameters have been presented in section 2.3.

We have described the statistical test method in section 2.

ð  This explanation is still insufficient. Please clarify which specific function in MATLAB was used, as there is no function named “Morlet Wavelet.” If you used the 'cwt' function, several other parameters are required, which should also be mentioned here.

The authors did not reply to the data rank issue raised in the previous round. Along with addressing the data rank issue, it is also important for study replication to specify whether you included the initial reference when re-referencing to the average.

Below is an excerpt from the previous round:

According to the EEGLAB official documentation, it is crucial to ensure that the data does not have rank deficiency before performing ICA. Please visit the documentation. The documentation highlights that incorrect re-referencing can cause rank deficiency and provides solutions to address this issue. Please explicitly mention this issue and explain how the authors addressed it in detail. I believe it’s likely that you used the PCA option, then you can just say “we performed ICA with the pca option to match the data rank to handle the rank-deficiency issue [a relevant paper]”, or you can say “We re-referenced to the average with the initial reference [a relevant paper], ensuring the full rank of the data”, as recommended including the initial reference for re-referencing.

https://eeglab.org/tutorials/06_RejectArtifacts/RunICA.html#issues-with-data-rank-deficiencies

https://sccn.ucsd.edu/wiki/Makoto's_preprocessing_pipeline#A_study_on_the_ghost_IC_was_published_.28added_on_04.2F04.2F2023.29

Comments on the Quality of English Language

The paper still needs to be proofread by native speakers. (e.g., “Electroencephalography (EEG) have been widely applied in the analysis of …”)

Author Response

  • Comments 1:

The way the authors cited [9] might be somewhat misleading. I recommend revising it to: "Therefore, EEG signals can be applied to decode kinematics and surface electromyography (EMG) during walking, as demonstrated in [9].".

Responses to the Comments 1:

Thanks for the constructive comments. We have revised the sentence you mentioned and highlighted in red.

  • Comments 2:

This explanation is still insufficient. Please clarify which specific function in MATLAB was used, as there is no function named “Morlet Wavelet.” If you used the 'cwt' function, several other parameters are required, which should also be mentioned here.

Responses to the Comments 2:

Thanks for the constructive comments. We did not used the function in MATLAB. The Morlet Wavelet function written by oneself based on the formula of Morlet wavelet.

  • Comments 3:

The authors did not reply to the data rank issue raised in the previous round. Along with addressing the data rank issue, it is also important for study replication to specify whether you included the initial reference when re-referencing to the average.

Below is an excerpt from the previous round:

According to the EEGLAB official documentation, it is crucial to ensure that the data does not have rank deficiency before performing ICA. Please visit the documentation. The documentation highlights that incorrect re-referencing can cause rank deficiency and provides solutions to address this issue. Please explicitly mention this issue and explain how the authors addressed it in detail. I believe it’s likely that you used the PCA option, then you can just say “we performed ICA with the pca option to match the data rank to handle the rank-deficiency issue [a relevant paper]”, or you can say “We re-referenced to the average with the initial reference [a relevant paper], ensuring the full rank of the data”, as recommended including the initial reference for re-referencing.

https://eeglab.org/tutorials/06_RejectArtifacts/RunICA.html#issues-with-data-rank-deficiencies

https://sccn.ucsd.edu/wiki/Makoto's_preprocessing_pipeline#A_study_on_the_ghost_IC_was_published_.28added_on_04.2F04.2F2023.29.

Responses to the Comments 3:

Thank you very much for your comment. However, we did not use EEGLAB when performing ICA. We have explained in section 2.2. The ocular correction ICA was conducted in Analyzer software, which is developed by Brain Products company(www.brainproducts.com).